# Multimodal Blockade of the Renin-Angiotensin System Is Safe and Is a Potential Cancer Treatment for Cats

**DOI:** 10.3390/vetsci9080411

**Published:** 2022-08-05

**Authors:** John S. Munday, Thomas Odom, Keren E. Dittmer, Sarah Wetzel, Katharina Hillmer, Swee T. Tan

**Affiliations:** 1School of Veterinary Science, Massey University, Palmerston North 4410, New Zealand; 2Family Vets Rotorua, Rotorua 3010, New Zealand; 3Gillies McIndoe Research Institute, Wellington 7184, New Zealand

**Keywords:** cat, cancer, renin-angiotensin system, oncology, therapy, cancer treatment

## Abstract

**Simple Summary:**

As activation of the renin-angiotensin system (RAS) promotes cancer cell growth, medications that inhibit RAS activation could reduce cancer progression. However, studies in people in which RAS has been inhibited by a single treatment have not been consistently beneficial, possibly as RAS can be activated by many different cellular pathways. Multiple treatments have been used to more consistently block RAS in people, but such multimodal treatments have never previously been evaluated in veterinary species. In the present study, the safety of multimodal RAS inhibition using a combination of five treatments was assessed in six cats with cancer. Cats were treated for 8 weeks and none of the cats developed low blood pressure, evidence of kidney or liver disease, or significant adverse effects. Of the six cats enrolled in the study, one cat was withdrawn from the study due to difficulties administering the medications and another cat died of an unrelated cause. Two cats were euthanatized due to cancer progression during the study period while two cats completed the 8-week treatment period. The study showed that a multimodal blockade of RAS has the potential to be a safe and cost-effective treatment for cancer in cats.

**Abstract:**

The role of the renin-angiotensin system (RAS) in cancer growth and progression is well recognized in humans. However, studies on RAS inhibition with a single agent have not shown consistent anticancer effects, potentially due to the neoplastic cells utilizing alternative pathways for RAS activation. To achieve more complete RAS inhibition, multimodal therapy with several medications that simultaneously block multiple steps in the RAS has been developed for use in humans. In the present study, the safety of multimodal RAS inhibition using atenolol, benazepril, metformin, curcumin, and meloxicam was assessed in six cats with squamous cell carcinomas. Cats were treated for 8 weeks, with blood pressure measured and blood sampled five times during the treatment period. None of the cats developed hypotension, azotemia, or increased serum liver enzyme concentrations. The packed cell volume of one cat decreased to just below the reference range during treatment. One cat was reported to have increased vomiting, although this occurred infrequently. One cat was withdrawn from the study due to difficulties administering the medications, and another cat died of an unrelated cause. Two cats were euthanatized during the study period due to cancer progression. Two cats completed the 8-week study period. One was subsequently euthanized due to cancer progression while the other cat is still alive 32 weeks after entering the study and is still receiving the multimodal blockade of the RAS. This is the first evaluation of multimodal blockade of the RAS in veterinary species. The study showed that the treatment is safe, with only mild adverse effects observed in two treated cats. Due to the small number of cats, the efficacy of treatment could not be evaluated. However, evidence from human studies suggests that a multimodal blockade of RAS could be a safe and cost-effective treatment option for cancer in cats.

## 1. Introduction

The role of the renin-angiotensin system (RAS) in controlling blood pressure and volume is well established. However, more recent evidence shows that RAS also influences cell growth and replication [1]. As illustrated in Figure 1, RAS is activated when prorenin is cleaved to form renin [2]. Renin converts angiotensinogen to angiotensin I, which is then rapidly converted by the angiotensin-converting enzyme (ACE) to angiotensin II. Angiotensin II activates the angiotensin II receptor 1 (AT1R) [3], which promotes cellular growth and migration [4].

The ability of RAS activation to influence cell behavior suggests activation of the RAS may be important in cancer development and progression. This is supported by the observation that people taking RAS inhibitors less frequently develop some types of cancer [5]. Additionally, increased AT1R expression by cancer cells is associated with reduced cell differentiation, increased invasiveness, increased angiogenesis, and, for some cancer types, reduced overall survival [6,7,8]. 

Due to the role of the RAS in promoting cancer progression, there has been much interest in the use of RAS inhibitors to treat human cancers. The majority of these studies used either an ACE inhibitor or an AT1R blocker. While some studies have shown promising results, other studies reported minimal efficacy of the treatment and there is currently no conclusive evidence that RAS inhibition using a single treatment slows cancer growth or progression [9,10]. One reason for the lack of consistency between studies could be the degree of redundancy within the RAS with multiple bypasses and converging pathways providing alternative means to generate angiotensin II. Neoplastic cells are well recognized as being able to utilize alternative pathways, so only blocking individual components of the RAS pathway may not be enough to prevent RAS activation. To counter this, a multimodal therapy to block multiple components of the RAS has been developed and this therapy was recently used to treat human patients with glioblastoma [11]. The multimodal therapy was developed using repurposed off-patent medications that are inexpensive and known to be well-tolerated and safe. 

The present report describes a similar multimodal blockade of the RAS as a potential treatment for squamous cell carcinoma (SCC) in cats. The medications used in the treatment included atenolol, which is a beta-blocker and inhibits the production of prorenin, Ref. [12] and meloxicam, which reduces the conversion of prorenin to renin by cyclooxygenase 2-mediated reduction in prorenin receptor number [13]. Conversion of prorenin to renin is also inhibited by curcumin, which inhibits the cathepsins, and by metformin, which reduces renin production by inhibiting insulin-like growth factor 1 [14,15]. Finally, the ACE inhibitor benazepril was included to prevent the conversion of angiotensin I to angiotensin II [9]. 

Prior to the present study, there have been few studies on RAS inhibition as a potential cancer treatment in cats. However, a recent study showed inhibition of AT1R reduced cell growth and invasion in an in vitro model of feline colon cancer [16]. Additionally, metformin was used as a single agent in a two-week-long study of 9 cats with tumors. While the small number of cats enrolled and the short length of the study prevented drawing firm conclusions about efficacy, the author concluded that metformin showed potential as a cancer treatment for cats [17]. In a study of 11 cats with urothelial carcinomas treated with meloxicam, the cats survived longer than expected, potentially due to RAS inhibition [18]. Meloxicam, in combination with other chemotherapeutics, has also been used in cats with mammary carcinomas, although whether meloxicam influenced survival time in these studies is uncertain [19].

The aim of the present study was to investigate the safety of a multimodal blockade of RAS in cats. To do this, six cats with incurable SCCs were carefully monitored for adverse treatment effects, as described in the Veterinary Cooperative Oncology Group—Common Terminology Criteria for Adverse Events (VCOG-CTCAE v2) guidelines [20]. The results show that multimodal RAS inhibition is safe and generally well tolerated. Efficacy of treatment was not able to be evaluated in the study due to the small number of treated cats. 

## 2. Materials and Methods

### 2.1. Study Design 

An 8-week-long, open-label, proof-of-concept phase 1 clinical trial was performed to investigate the safety of a multimodal blockade of RAS in six cats. Cats received half doses of each medication during week 1 of the study (Table 1), and then full doses from week 2 onwards. Cats received 50 mg metformin hydrochloride once every 24 h (q24hr; Metformin 500 mg, Apotex NZ Ltd., Auckland, New Zealand), 12.5 mg atenolol q24hr (Mylan atenolol 50 mg, Mylan NZ Ltd., Auckland New Zealand), 2 mg benazepril hydrochloride q24hr (Apex benazepril 5 mg/mL oral solution, Dechra Veterinary Products, Somersby, New South Wales, Australia), 0.2 mg meloxicam q24hr (Ilium meloxicam 0.5 mg/mL oral suspension for cats, Troy Laboratories Pty Ltd., Glendenning New South Wales, Australia), and 160 mg curcumin q12hr (NHV turmeric 160–180 mg curcumin/mL dietary supplement, NHV Natural Pet Products, Vancouver, BC, Canada). All medications were given orally, with the metformin and atenolol administered as crushed tablets and the other three medications given as liquids using a syringe. 

### 2.2. Animals and Eligibility Criteria 

All cats in the study were client-owner animals. To be eligible, the cats had to have been diagnosed with an SCC either by cytology or histology. In addition, the SCCs had to be classified as incurable. Four SCCs were considered incurable due to invasion into surrounding tissue, while surgical excision of SCCs on the eyelids of two cats (cats 3 and 6) required enucleation, which was declined by the owner. Cats were excluded if they showed clinical signs of ill-health other than due to their SCCs, if their systolic blood pressure was below 120 mmHg, or if complete blood count (CBC) and serum biochemistry at the time of initial assessment revealed evidence of underlying disease. End-of-life decisions were made by clients in consultation with their veterinarian as normal. 

### 2.3. Monitoring

Clinical examination, assessment of the SCC, oscillometric blood pressure measurement, CBC (IDEXX laboratories Ltd., Palmerston North, New Zealand), and a serum biochemistry panel (feline geriatric panel, IDEXX Laboratories Ltd.) were performed as described in Table 2. Due to the fractious nature of Cat 1, sedation using 0.01 mg/kg medetomidine and 0.2 mg/kg butorphanol via IM injection was required when blood was drawn for monitoring at weeks 1, 2, and 4 of the study. During the revisits for blood sampling, the owners of the cats also completed a survey in which they described their perception of their cat’s quality of life. Owners were also asked to record any changes in appetite, vomiting, or diarrhea, as well as for comments regarding the administration of the treatment. 

When cats were euthanatized, the date and cause of death were recorded. A full necropsy examination was performed with samples of the neoplasm, draining lymph nodes, and any other abnormalities were taken for histology. In addition, samples of the lung, heart, liver, stomach, small intestine, colon, kidney, pancreas, spleen, and bladder were also collected for histologic examination to assess for potential toxicity due to the treatment. 

## 3. Results

Seven cats were initially recruited into the study. However, one of these cats was excluded as baseline serum biochemistry revealed mild azotemia consistent with underlying kidney disease. As summarized in Table 3, three cats had oral SCCs, two had eyelid SCCs (including one cat with bilateral eyelid SCCs with the SCC on one side recurring after enucleation), and one cat had an SCC on the external nose. Two cats were euthanatized due to SCC progression prior to completing the 8 weeks of the study: one because the SCC invaded the rostral parts of both mandibles preventing the cat from eating (cat 4); while the other as the external nose SCC began to interfere with normal breathing (cat 2). One cat with an oral SCC was euthanatized after developing an aortic thromboembolism (cat 5). Post-mortem examination also revealed hypertrophic cardiomyopathy and the death of this cat was considered to be unrelated to both the SCC and the treatment. One cat was withdrawn from the study after 4 weeks because the owner was unable to consistently administer the treatment (cat 1). This cat was unusually fractious and was the only cat in the study that required sedation for blood sampling. Both of the cats that had eyelid SCCs completed the 8-week study period. Treatment was continued for a further 2 weeks in one of these cats before the owner stopping giving the atenolol and metformin due to difficulties in administering the crushed tablets. The cat continued to receive the three liquid medications for a further 15 weeks until it was euthanatized due to SCC progression (cat 3). The other cat with an eyelid SCC has continued with the multimodal treatment for 32 weeks, and the cat remains healthy with little observable progression of the SCC (cat 6). 

Monitoring of the cats throughout the study revealed that both the weight and blood pressure of all the cats remained roughly constant throughout the study (Table 4). Furthermore, the hematocrit of 5 of 6 cats remained stable, with only one cat developing a marginally low hematocrit after four weeks of treatment. This mild anemia was consistent with a grade 1 adverse effect that was possibly attributed to the treatment according to the Veterinary Cooperative Oncology Group—Common Terminology Criteria for Adverse Events (VCOG-CTCAE v2) guidelines [20]. Increased urea or creatinine concentrations were not seen in any of the cats. Serum alanine aminotransferase (ALT) activity was increased in one sample during the study. However, the sample was hemolyzed and the result was considered artifactual. Further supporting a spurious result, the cat showed no other evidence of liver or muscle disease during the study, and serum ALT activity was within normal limits when the next sample was taken two weeks later. 

All owners reported an improvement in the quality of life of their cats in the week following initiation of the treatment, and none reported a negative impact of the medication within the 8-week study period. One cat (cat 3) was reported to vomit around twice a week while receiving the medication. Although the cat regularly vomited prior to being included in the study, the owners believed the cat was vomiting more frequently when receiving the multimodal treatment. The increased vomiting was not considered significant enough by the owners to withdraw the cat from the study and the cat did not lose weight during the study. The increased vomiting was consistent with a grade one adverse effect that was possibly attributed to the treatment as defined by the VCOG-CTCAE v2 guidelines [20].

All owners commented that the daily administration of five oral medications was difficult. However, five owners were able to continue dosing their cats throughout the study and only one cat had to withdraw from the study due to difficulties administering the medications. 

Samples of the SCCs taken from the four cats that were euthanatized during or after completing the study were examined histologically. As expected, the neoplasms from the oral cavity and nasal planum appeared as invasive nests and trabeculae of poorly differentiated keratinocytes. Interestingly, the oral and nasal planum neoplasms contained mitotic rates that ranged from 2 to 6 mitotic figures/10 hpfs, which was considered unusually low for neoplasms of this type. The bilateral eyelid SCCs that were present on cat 3 had both spread to the surrounding haired skin, but both neoplasms remained predominantly confined to the epidermis and more superficial aspects of the dermis. Tumor metastases to local lymph nodes were not observed in any of the cats. The sections of other organs from these cats were within normal histological limits.

## 4. Discussion 

When assessing a new treatment for cancer, it is important to confirm that the treatment is safe and does not result in adverse effects that significantly reduce the quality of life of the patient. While all five of the medications used in the study have individually been studied in cats, they had not previously been used together, so the primary goal of the study was to assess the safety of multimodal therapy. Due to the key role of RAS in maintaining blood pressure, hypotension was considered the most likely adverse effect of the treatment. However, none of the cats developed clinical signs of hypotension, and systolic blood pressure measurements remained above 120 mmHg in all six treated cats. Indeed, the blood pressure measurements in several of the cats suggested hypertension, although this may have been due to anxiety caused by repeated visits to the veterinary clinic [21]. While infrequent vomiting was observed in one of the treated cats, the cat did not have a reduced appetite or lose weight, and the vomiting was not considered by the owner of the cat to be frequent enough to discontinue treatment. Overall, the results of the present study showed that a multimodal blockade of RAS is safe in cats, with adverse effects infrequent and mild. 

As chronic kidney disease is common in older cats, the possibility that RAS inhibition could reduce renal blood flow and worsen subclinical kidney disease was considered. However, as none of the cats developed increased serum urea or creatinine concentrations, this suggests that RAS inhibition did not significantly impact kidney function in these cats. Metformin has previously been associated with increased liver enzyme activity and anemia in cats [17,22]. In contrast, liver enzyme activities were not increased in the present study. While the PCV of one cat decreased during treatment, the PCV was only slightly below the reference range. As the cat was euthanatized shortly after the blood sample was taken, is unknown whether the cat would have developed more significant anemia or whether the PCV would have returned to normal levels. No decrease in the PCV was observed in any of the other five treated cats. 

The dose rates used in the present study were derived using clinical experience and from published literature. The meloxicam dose used in the study is the dose that is routinely used by clinicians at the Massey Veterinary Teaching Hospital for long-term pain relief in cats. The dose rate of curcumin was around 40 mg/kg q12hr as recommended by the manufacturer of the supplement. However, absorption of curcumin is poor and, although the supplement contained piperine, which greatly increases bioavailability [23], the proportion of curcumin absorbed was unknown. In comparison, a dose of 12.5 mg/kg q12hr curcumin was used in multimodal RAS blockade therapy used in humans [11]. Curcumin has a high safety margin and dose rates of 120 mg/kg are well-tolerated in people [23]. The dose rates of benazepril and atenolol were those previously reported to be safe in cats [24,25]. There have been two previous studies of metformin in cats. Significant vomiting and anorexia were reported in both studies, in which dose rates of 10 mg/kg q12hrs [17] and 50 mg/cat q12hrs [22] were used. As twice-daily dosing caused significant side effects in both previous studies, a dose rate of 50 mg/cat q24hrs was used in the present study. This avoided significant side effects due to metformin but, due to the half-life of metformin in cats [22], likely resulted in cats having low blood metformin concentrations in the hours prior to the next treatment. However, while these low levels would be undesirable if using the medication to treat diabetes, it is less certain whether transient low metformin blood concentrations significantly impact the ability of the multimodal therapy to inhibit RAS activation. 

Another important factor to consider when evaluating a novel cancer treatment for cats is the ease of administration of the medication. Ideally, the medication should be palatable, dosing should be infrequent, and cats should be dosed in their normal environment. In the present study, all medications except curcumin were given just once a day. The formulations of meloxicam, benazepril, and curcumin that were used are palatable liquids designed for easy administration to cats. In contrast, both metformin and atenolol were only available as tablets formulated for humans. These tablets contained much higher doses than required for cats and had to be crushed prior to administration. Furthermore, as the cats appeared to find the taste of the crushed tablets unpleasant, cats were sometimes reluctant to eat food containing the medication. Due to this aversion to medicated food, some owners found it easier to dissolve the crushed tablets in water and administer the medication via a syringe. All owners reported that the daily dosing was difficult over an extended period and dosing became so problematic for one owner that the cat was withdrawn from the study. The difficulties in dosing the cats in the present study suggest that, if multimodal RAS inhibition is investigated in a larger number of cats, it would be beneficial to combine all medications into a single palatable daily dose. If simple daily dosing can be achieved, a multimodal blockade of RAS is likely to be easier and safer than dosing with cytotoxic or radiation-based cancer therapies. 

The likely cost of a novel cancer treatment is also an important consideration with many current options for treating cancer in cats too expensive to be frequently used. A significant advantage of multimodal RAS inhibition is that the components of the therapy are repurposed off-patent, inexpensive drugs. Multimodal RAS inhibition, therefore, has the potential to be a highly cost-effective way to treat cancer in cats. The difference in cost of using a newly developed cancer therapeutic compared to using repurposed medications is especially apparent when treating human cancers. Showing the efficacy of a multimodal blockade of RAS with repurposed medications for feline cancers will further support the use of this treatment approach in humans. 

The final factor to be considered when assessing the likely success of a multimodal blockade of the RAS as a novel cancer treatment for cats is the efficacy of the treatment in slowing cancer growth and progression. While the study was able to confirm that the use of a multimodal blockade of the RAS did not result in significant adverse effects in treated cats, it remains uncertain if the treatment prolonged life. While the unexpectedly low mitotic rate within the SCCs provided tantalizing evidence of a possible benefit, additional studies of large numbers of cats are required to evaluate the efficacy of the treatment. It should be noted that all owners reported that the treatment improved the quality of life of their cats. Whether this was due to slowed cancer growth, pain relief from meloxicam, or a placebo effect in this open-label study, remains uncertain. However, owners of both cats that completed the 8-week study opted to continue the treatment because they felt it had reduced cancer growth and progression. 

Cats with SCCs were chosen for the study population for two reasons. Firstly, there is evidence that human SCCs respond to treatment using RAS inhibitors [26,27]. Secondly, SCCs are very common cancers in cats and it was hoped that a sufficient number of cats with oral SCCs could be enrolled in the study. However, recruiting cats with oral SCCs was problematic, as many cats presented with advanced disease that resulted in euthanasia at, or shortly after, the time of diagnosis. The lack of a sufficient number of cats with oral SCCs necessitated the inclusion of cats with SCCs on haired skin. Including these cats had the advantages of being able to quickly recruit enough cats, as well as allowing changes in size or appearance of the cancers to be more easily observed. However, cats with oral SCCs have average survival times of just 4 to 6 weeks [28,29] while cats with SCCs of haired skin often survive a year after diagnosis [30]. Therefore, the high variability of the expected survival times of cats included in the study further decreased the ability of the study to detect the efficacy of the treatment. 

A disadvantage of the study was the reliance on owners to administer the treatment. While all six owners were highly committed to the study, it cannot be excluded that some doses could have been missed, either because the cats were able to avoid ingesting the treatment or because some treatments were accidentally not given. Ideally, repeated blood samples would have been taken to measure the concentrations of each treatment in the blood; however, this was not feasible in this pilot study. In the future, additional work will be required to confirm the optimal doses of each of the medications used in the multimodal blockade of the RAS.

## 5. Conclusions

This is the first time that multimodal RAS inhibition has been used in veterinary medicine. The observations suggest that the treatment is safe with infrequent and mild adverse effects. Daily oral dosing of multiple medications was difficult. Whether or not RAS inhibition slowed cancer growth and progression could not be determined, although some evidence suggested a possible benefit of the treatment. A multimodal blockade of the RAS, therefore, represents a potentially safe and cost-effective option to treat cancer in cats. 

## Figures and Tables

**Figure 1 vetsci-09-00411-f001:**
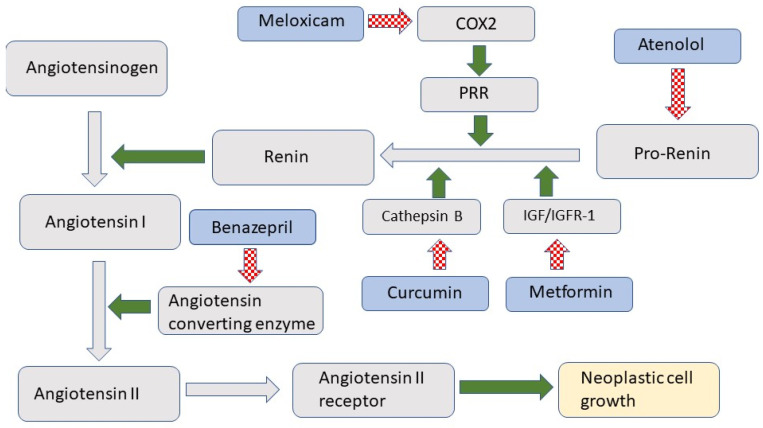
Schematic diagram of the renin-angiotensin system (RAS). RAS components are indicated in grey while treatments used in the study are indicated by a blue shape. Green arrows indicate promotion while hashed red arrows indicate inhibition. COX2 is cyclooxygenase 2; PRR is prorenin receptor; IGF/IGFR-1 is insulin-like growth factor/insulin-like growth factor receptor 1.

**Table 1 vetsci-09-00411-t001:** Treatments used for multimodal renin-angiotensin system inhibition. * Due to variability in the natural product used to administer curcumin, the full dosage is expected to be in the range of 160–180 mg.

	Metformin	Benazepril	Meloxicam	Atenolol	Curcumin *
1–7 days	25 mg	1 mg	0.1 mg	6.25 mg	80 mg
8 days to completion	50 mg	2 mg	0.2 mg	12.5 mg	160 mg

**Table 2 vetsci-09-00411-t002:** Summary of treatments and monitoring throughout the 8-week study. CBC is complete blood count.

Day 0	Baseline Clinical Examination, Blood Pressure Measurement, CBC, and Serum Biochemistry
Days 0–6	Cats receive half doses of all medications
Day 7	Clinical examination, blood pressure measurement, CBC, and serum biochemistry
Day 7–13	Cats start on full doses of all medications
Day 14	Clinical examination, blood pressure measurement, CBC, and serum biochemistry
Day 14–27	Continue on full doses of medications
Day 28	Clinical examination, blood pressure measurement, CBC, and serum biochemistry
Day 28–41	Continue on full doses of medications
Day 42	Clinical examination, blood pressure measurement, CBC, and serum biochemistry
Day 42–55	Continue on full doses of medications
Day 56	Clinical examination, blood pressure measurement, CBC, and serum biochemistry
Day 56-	Optional continuing on full or partial medications

**Table 3 vetsci-09-00411-t003:** Summary of the clinical information and clinical outcome of cats receiving a multimodal blockade of the renin-angiotensin system. * indicates that, at the time of writing, the cat is still alive and receiving multimodal treatment. FS indicates a female spayed cat; MC indicates a male castrated cat.

	Age/Sex	Location of Squamous Cell Carcinoma	Time in Trial (Weeks)	Outcome
Cat 1	10/FS	Oral	4	Withdrew due to difficulties in administering the medications
Cat 2	10/FS	External nose	5	Cat euthanatized due to cancer progression
Cat 3	9/FS	Bilateral eyelid	10	Full multimodal therapy given for 10 weeks then only meloxicam, benazepril, and curcumin. Cat euthanatized due to cancer progression 25 weeks after enrolment in study
Cat 4	14/MC	Oral	5	Cat euthanatized due to cancer progression
Cat 5	13/MC	Oral	6	Cat euthanatized after developing aortic thromboembolism
Cat 6	11/FS	Eyelid	32 *	No evidence of cancer progression after receiving multimodal therapy for 32 weeks

**Table 4 vetsci-09-00411-t004:** Weight, systolic blood pressure, and select results of blood testing of cats receiving a multimodal blockade of the renin-angiotensin system. ALT is alanine aminotransferase activity (IU/L), PCV is the packed cell volume (%), urea is serum urea concentration (mmol/L). Values in brackets are the reference ranges as supplied by the veterinary diagnostic laboratory. * The sample contained hemolysis and this value is probably artefactually elevated.

	Weight	Systolic Blood Pressure (mm/Hg)	Urea (5.7–12.9)	ALT (0–100)	PCV% (0.24–0.45)
**Cat 1**					
Baseline	3.92	194	11.1	45	0.31
Week 1	3.96	177	11.5	47	0.31
Week 2	3.95	157	11.4	55	0.28
Week 4	3.95	135	9.3	53	0.35
**Cat 2**					
Baseline	5.4	125	9.3	69	0.34
Week 1	5.43	166	10.1	47	0.32
Week 2	5.36	167	10.2	36	0.25
Week 4	5.27	155	9.1	41	0.32
**Cat 3**					
Baseline	4.01	187	9.7	45	0.44
Week 1	4.09	170	9.5	51	0.43
Week 2	4.67	209	9.8	56	0.46
Week 4	4.14	190	9.0	49	0.46
Week 6	4.1	210	7.7	48	0.49
Week 8	3.99	210	10.1	44	0.46
**Cat 4**					
Baseline	3.85	160	NA	NA	NA
Week 1	3.85	160	7.9	32	0.50
Week 2	3.85	180	7.6	35	0.47
Week 4	3.84	184	8.8	30	0.40
**Cat 5**					
Baseline	5.1	201	12.2	60	0.32
Week 1	5.16	164	12.0	55	0.32
Week 2	5.08	185	12.9	69	0.24
Week 4	5.06	153	11.0	89	0.23
**Cat 6**					
Baseline	6.44	150	8.4	62	0.47
Week 1	6.47	240	10.8	160 *	0.46
Week 2	6.47	164	10.6	86	0.50
Week 4	6.5	175	11.9	66	0.43
Week 6	6.36	140	11.5	45	0.46
Week 8	6.39	143	9.2	56	0.50

## Data Availability

The data presented in this study are available on request from the corresponding author.

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
