# Peer review of "Multimodal Blockade of the Renin-Angiotensin System Is Safe and Is a Potential Cancer Treatment for Cats"

_vetsci, 2022, doi:10.3390/vetsci9080411_

Round 1
Reviewer 1 Report
The manuscript submitted by Munday et al. entitled " Multimodal blockade of the renin-angiotensin system is safe and is a potential cancer treatment for cats" aims to evaluate the utility of the multimodal RAS inhibition in feline medicine, by its application in cats with squamous cell carcinoma. As the authors point out this study has several limitations. Indeed, the number of cats enrolled in this study is very low, mainly because the animals presented tumors with different locations and stages. Keeping in mind these limitations and the fact that the use of RAS inhibition has low application in human oncology, the rationale for using this type of therapy in veterinary medicine is also limited.
However, this manuscript reports the use of multimodalRAS inhibition therapy for the first time in veterinary, suggesting that this type of treatment is safe with infrequent and mild adverse effects in cats with SCC.
The introduction should be reorganized in order to improve the flow of the information given to the reader.
Author Response
The manuscript submitted by Munday et al. entitled " Multimodal blockade of the renin-angiotensin system is safe and is a potential cancer treatment for cats" aims to evaluate the utility of the multimodal RAS inhibition in feline medicine, by its application in cats with squamous cell carcinoma. As the authors point out this study has several limitations. Indeed, the number of cats enrolled in this study is very low, mainly because the animals presented tumors with different locations and stages. Keeping in mind these limitations and the fact that the use of RAS inhibition has low application in human oncology, the rationale for using this type of therapy in veterinary medicine is also limited.
However, this manuscript reports the use of multimodalRAS inhibition therapy for the first time in veterinary, suggesting that this type of treatment is safe with infrequent and mild adverse effects in cats with SCC.
Thank you for the positive feedback.
The introduction should be reorganized in order to improve the flow of the information given to the reader.
The introduction has been extensively modified and re-ordered to make the information clearer for the reader. In addition, Figure 1 has been added to clarify the RAS system and the medications used. The inclusion of this figure has allowed much of the information about the RAS system to be removed improving the flow of the introduction (lines 45-76).
Reviewer 2 Report
In general, this work is really interesting and well written. My comments aim to increase the scientific soundness and clarity of it.
Line 73 - please explain what ATI and ATII stand for. Line 85 - Please add a brief description of methodology applied in the study. Line 91 – Please provide any details concerning study animals (age, weight, sex, general health status etc.). Without these date the study design is not complete. Line 91 – it is not clear how many cats were used in the study. Lines 95-96 – Please explain what q24hr and q24 mean. Line 159 – according to veterinary anatomical nomenclature nasal planum is only part of external nose. Are the authors sure about this location? Maybe it would be more appropriate to use wider term “external nose” instead. Line 159 – Photographic documentation of squamous cells carcinoma locations is needed. Line 167-168 – Please explain what BUN and ALT stand for. Line 334 – The authors should check the format of references (seems that issue numbers and DOI links are missing).
Author Response
In general, this work is really interesting and well written. My comments aim to increase the scientific soundness and clarity of it.
Thank you for the positive feedback.
Line 73 - please explain what ATI and ATII stand for.
These abbreviations have been removed to improve clarity and readability (line 84)
Line 85 - Please add a brief description of methodology applied in the study.
This has been added as suggested (line 94)
Line 91 – Please provide any details concerning study animals (age, weight, sex, general health status etc.). Without these date the study design is not complete.
The age and sex of the cats has been added to Table 3 (line 177) while the weights of the cats is in Table 4 (line 195). The requirement that all cats be judged as being in good health at the start of the study has been added (line 128)
Line 91 – it is not clear how many cats were used in the study.
This information has been more clearly stated (line 107)
Lines 95-96 – Please explain what q24hr and q24 mean.
This has been added (line 110)
Line 159 – according to veterinary anatomical nomenclature nasal planum is only part of external nose. Are the authors sure about this location? Maybe it would be more appropriate to use wider term “external nose” instead.
This has been changed as suggested (line 160)
Line 159 – Photographic documentation of squamous cells carcinoma locations is needed.
While all SCCs were photographed during the study the authors do not feel that adding these photographs to the manuscript would be desirable. This is because the major focus of the study is the safety of the treatment rather than the cancers themselves. The study was not designed to determining efficacy and therefore detail about the SCCs in each cat was deliberately kept brief to ensure the focus of the manuscript remains on the safety of the treatment given.
Line 167-168 – Please explain what BUN and ALT stand for.
This has been added as suggested (lines 185-186)
Line 334 – The authors should check the format of references (seems that issue numbers and DOI links are missing).
The references have been checked. Many online only journals do not include issue numbers which is why not all references have these. DOIs were not included as the journal reference format does not include this information.
Reviewer 3 Report
The work entitled “Multimodal blockade of the renin-angiotensin system is safe and is a potential cancer treatment for cats” is an interesting research about a new anti-neoplastic treatment scheme in cats. This is the innovative clinical work on cats with SCC, and I recommend it for publication. However, before final decision please find my comments below.
Introduction;
Please consider a shorter description of RAS mechanisms and including the scheme instead. Please indicate the points of possible blockade on this scheme.
In this part lack the short description of safety assessment of anti-neoplastic drugs. Which indicators are taken into account? Please add some references?
Results
Lines138-158; The way the results were described shows that no cat survived the scheduled treatment cycle. Do I understand the Authors' intentions well?
Lines 164-167; should be moved to introduction.
I am concerned about the final low number of animals treated
Line 300: Limitations: The small number of animals under treatment is a work limitation.
Author Response
The work entitled “Multimodal blockade of the renin-angiotensin system is safe and is a potential cancer treatment for cats” is an interesting research about a new anti-neoplastic treatment scheme in cats. This is the innovative clinical work on cats with SCC, and I recommend it for publication. However, before final decision please find my comments below.
Thank you for the positive feedback.
Introduction;
Please consider a shorter description of RAS mechanisms and including the scheme instead. Please indicate the points of possible blockade on this scheme.
The authors consider this an excellent idea and figure 1 now is included to summarize the RAS and to indicate how the treatments work. The inclusion of figure one has allowed the introduction to be shortened and simplified (line 45-76)
In this part lack the short description of safety assessment of anti-neoplastic drugs. Which indicators are taken into account? Please add some references?
This information has been added as suggested (line 98-100)
Results
Lines138-158; The way the results were described shows that no cat survived the scheduled treatment cycle. Do I understand the Authors' intentions well?
4 of the 6 cats did not complete the 8-week experimental period, 2 cats did complete the 8 weeks with one cat still be alive. This was obviously not clearly stated and so the results and table 3 have been modified to make this information more clear (line 166-173 and 174-178)
Lines 164-167; should be moved to introduction.
This has been moved as suggested (line 99)
I am concerned about the final low number of animals treated
Line 300: Limitations: The small number of animals under treatment is a work limitation.
The authors agree that the number of animals is low and the low numbers mean that no conclusions can be made about treatment efficacy. However, the authors feel the study is still valuable as the number of cats included allow for confidence regarding the safety of the treatment.